

# On the dynamic nature of hydrological similarity

Ralf Loritz[1], Hoshin Gupta[2], Conrad Jackisch[1], Martijn Westhoff[3], Axel Kleidon[4], Uwe Ehret[1] and Erwin Zehe[1]

5  [1] Karlsruhe Institute of Technology (KIT), Institute of Water and River Basin Management, Karlsruhe, Germany

[2] University of Arizona, Department of Hydrology and Atmospheric Sciences; Tucson, AZ;

[3] Vrije Universiteit, Department of Earth Science, Amsterdam, Netherlands

[4] Max-Planck-Institut für Biogeochemie, Jena, Germany

10  *Correspondence to*: Ralf Loritz (Ralf.Loritz@kit.edu)

Key words: hydrological similarity, information theory, thermodynamics, functional units, landscape organization, data compression

**Abstract:** The increasing diversity and resolution of spatially distributed data on terrestrial systems greatly enhances the potential of hydrological modeling. Optimal and parsimonious use of these data sources implies, however, that we better understand a) which system characteristics exert primary controls on hydrological dynamics and b) to what level of detail do those characteristics need to be represented in a model.

In this study we develop and test an approach to explore these questions that draws upon information theoretic and thermodynamic reasoning, using spatially distributed topographic information as a straightforward example. Specifically, we subdivide a meso-scale catchment into 105 hillslopes and represent each by a two dimensional numerical hillslope model. These hillslope models differ exclusively with respect to topography related parameters derived from a digital elevation model; the remaining setup and meteorological forcing for each are identical. We analyze the degree of similarity of simulated discharge and storage among the hillslopes as a function of time by examining the Shannon information entropy. We furthermore derive a 'compressed' catchment model by clustering the hillslope models into functional groups of similar runoff generation using normalized mutual information as a distance measure.

Our results reveal that, within our given model environment, only a portion of the entire amount of topographic information stored within a digital elevation model is relevant for the simulation of distributed runoff and storage dynamics. This manifests through a possible compression of the model ensemble from the entire set of 105 hillslopes to only 6 hillslopes, each representing a different 'functional



group', which leads to no substantial loss in model performance. Importantly, we find that the concept of hydrological similarity is not necessarily time-invariant. On the contrary, the Shannon entropy as measure for diversity in the simulation ensemble shows a distinct annual pattern, with periods of highly redundant simulations, reflecting coherent and organized dynamics, and periods where hillslopes operate in distinctly

different ways.

We conclude that the proposed approach provides a powerful framework for understanding and diagnosing how and when process organization and functional similarity of hydrological systems emerges in time. Our approach is neither restricted to the model, nor to model targets or the data source we selected in this study. Overall, we propose that the concepts of hydrological systems acting similarly (and thus

giving rise to redundancy) or displaying unique functionality (and thus being irreplaceable) are not mutually exclusive. They are in fact of complementary nature, and systems operate by gradually changing to different levels of organization in time.



# 1. Introduction

## 1.1 Motivation

[1]   This paper addresses the question "*How important is spatial variability of terrestrial system characteristics and meteorological forcing when viewed from the perspective of stream flow generation and distributed water storage?*" While this question has motivated hydrologists since the early days of our science, it gained substantial attention with the development of distributed hydrological models, and it seems fair to say that attempts to address the question still lie at the heart of every distributed model application *(Beven, 1989; Freeze and Harlan, 1969; Refsgaard, 1997; Hrachowitz and Clark, 2017)*.

[2]   Needless to say, this question has not found easy answers. Besides the lack of sufficient process understanding (in part due to the difficulty of gathering relevant data about hydrologic systems), there is also the uncertainty we unavoidably encounter when dealing with the steadily growing and changing pool of geo-information *(Musa et al., 2015)*. For instance land surface digital elevation information is now available at a resolution of 25 m globally *(Farr et al., 2007)*. Similarly, weather radar coverage is available for large parts of Europe, providing accumulated 15 min precipitation estimates at 4 km resolution *(Huuskonen et al., 2014)*. Despite the huge potential for model improvement provided by these new and diverse pools of information, a danger associated with their use is that we can "*miss the forest for the trees*" unless we are able to determine which information contained in the data is of relevance to the questions we seek to answer.

[3]   We therefore now face the problem of how to discriminate important details about the hydrological landscapes from idiosyncratic ones, and hence must deal with the challenge of how to identify which characteristics explain hydrological similarity *(Blöschl and Sivapalan, 1995)*. This study is largely motivated by the "power" view introduced by *Wagener and Gupta (2005)* which advocates "*a need to develop better methods for characterizing and extracting relevant information from data*" (see also *Gupta and Nearing, 2014*). Our specific objective is to propose an approach addressing this issue, by drawing upon an information theoretic perspective to extract and quantify the relevant information for spatially distributed hydrological modeling, and by using thermodynamic reasoning to explain why only a portion of the full information content available in the data is relevant.

## 1.2 Background

[4]   From a thermodynamic perspective, streamflow generation is driven by differences in potential energy between the upslope catchment areas and the stream channel. Some portion of this available energy is dissipated during runoff concentration, while the remaining part is exported from the catchment as the kinetic energy of streamflow *(Kleidon et al., 2013)*. These potential energy differences depend





largely on catchment topography, and on the space-time patterns of precipitation (*Zehe et al., 2013*). Accordingly, we might be naturally drawn to expect that large spatial variations in both characteristics will result in large spatial variations in runoff generation. However, when exactly should spatial variation be considered "large" enough that we need to explicitly account for it?

[5]    In the context of spatially distributed rainfall, this latter question has received considerable attention (e.g. *Obled et al., 1994; Arnaud et al., 2002; Tetzlaff et al., 2005; Zehe et al., 2005; Das et al., 2008)*. In general, the predominant view that seems to emerge from these studies is that the impact (on runoff simulations) of spatial distribution in rainfall increases with size of the area considered. This is often traced back to the growing importance of flood routing, in combination with the average spatial extent of

typical rain storms (e.g. *Smith et al., 2004; Lobligeois et al., 2014*). Nevertheless, no consensus has yet emerged as to whether this statement is generally valid, and no guidelines exist regarding under which conditions the use of information regarding the spatially distributed nature of rainfall becomes inevitable (*Emmanuel et al., 2015*).

Similarly, the question of how strongly the spatial resolution of a DEM affects the results of a distributed

model application has been investigated in various studies (e.g. *Schoorl et al., 2000; Thompson et al., 2001; Sørensen and Seibert, 2007*). For instance *Zhang and Montgomery (1994)* varied the resolution of their DEM and reported that spatial resolutions finer than 10 m did not result in significant improvements to the simulation results of their hydrological model. *Chaubey et al. (2005)* tested the influence of DEM spatial resolution on simulation results of the Soil Water and Assessment Tool (SWAT) and reported that

grid size has a significant influence on different watershed responses, as well as on the sub-basin classification implemented in SWAT. However, as with the case of distributed rainfall, the results of these studies do not point to a generic approach, nor to any general conclusions regarding the importance of DEM-resolution for distributed hydrological modeling.

[6]    Overall, this lack of a coherent image certainly reflects the varying sensitivities of different model

structures (*Das et al., 2008*), the dependence on scope and scale of the model exercise *(Blöschl and Sivapalan, 1995*) and on differences among hydrological landscapes (*Beven, 2000*). It seems, therefore, that an investigation of the role of distributed information in hydrological modeling may benefit from a more generic and systematic approach, one that may be generalized to different spatially distributed data sources and models, and that is able to cope with interactions among them in a straightforward manner. In

contrast to much of the aforementioned work, which has relied primarily on statistical methods, the purpose of the work reported here is to investigate the extent to which information theory *(Cover and Thomas, 2005*) is able to provide instructive measures that are suitable for this purpose.




[7]    More specifically the main objective of this study is to present and test an approach to quantify the relevance of spatially distributed data sources for hydrological simulations drawing from information theory. We exemplify this approach using catchment topography as distributed information source as well as stream flow and soil water storage as modeling targets, however, the general mindset of the approach is applicable to any distributed information source such as spatially distributed rainfall or geology as well as to a wide range of arbitrary model target and different distributed models.

## 1.3 The role of surface topography in hydrological modeling

[8]    Despite the fact that DEM's provide the basis for identifying watershed boundaries, river networks and potential energy differences in the landscape, several studies have concluded that topography alone is a weak descriptor for inferring similarity in hydrological behavior. For instance, *Zehe et al. (2005)* showed that the topographic wetness index (*Beven and Kirkby, 1979*), a popular topographic similarity measure, failed to explain soil moisture variability and similarity in runoff generation in a lower mesoscale catchment. *Fenicia et al. (2016)* and *Jackisch (2015)* showed that topography alone might be a poor guide for subdividing a $256 \ km^2$ catchment into different functional units, and questioned the explanatory power of the topography in this respect. Our own work, *Loritz et al. (2017),* has shown that an "effective" representation of two different catchments by a single representative hillslope was able to provide successful simulations of their inter-annual runoff responses and annual storage dynamics. Together, these findings suggest that an informationally "compressed" representation of the topographic map may be able to preserve the relevant information regarding geopotential differences that drive runoff generation.

[9]    In line with these findings, we therefore pose the hypothesis that "*although a highly-resolved DEM contains a large amount of information about topography, not all of this spatially distributed information is relevant for the generation of hydrological predictions*". Following *Weijs et al. (2013)*, it seems reasonable that information theory may provide a natural framework for dealing with such compression of information in hydrologic science. The term "compression" was originally coined by Claude Shannon to refer to the quantification, storage and communication of information (*Shannon, 1948*). In environmental science, information-theoretic concepts such as the "Shannon entropy" have found widespread use in various applications (e.g. *Brunsell, 2010; Weijs et al., 2013a; Yakirevich et al., 2013*), ranging from uncertainty assessment in 3-D geological models (*Schweizer et al., 2017*) to the delineation of water resource zones in Japan (*Kawachi et al., 2001*). For an introduction to, and detailed review of, information theoretic concepts we refer the reader to *Cover and Thomas (2005), Singh (2013),* and *Weijs and van de Giesen (2013)*.

[10] With respect to the above finding it is important to note that compressibility relates to order or organization (*Davies, 1990*). The identification of relevant information within distributed system





characteristics is therefore closely linked to the identification of spatial organization and thus with the identification of hydrological similar functioning areas (*Sivapalan, 2005*). As pointed out by *Zehe et al. (2014)*, these "functional units" may be straightforwardly defined in thermodynamic terms as any flux is driven by a specific gradient while it performs work against a specific flow resistance. Similarity of both, the relevant drivers and the resistance terms is a sufficient criterion to expect that two systems behave similarly with respect to the generation of a flow, and with regard to the associated entropy production.

[11]  If we transfer this concept to runoff generation, differences in the geopotential (topography) act as driver since runoff is driven by gravity and feeds from free water. The resistance term, on the other hand depends either on surface roughness (and thus for instance on the vegetation in case of overland flow) or on the pattern of subsurface conductance and apparent preferential pathways. Yet, the gradient flux-resistance relation is non-unique, because a twice as large driver in combination with a twice as large resistance results in exactly the same flux. It is this non-uniqueness, which explains why two hillslopes with distinctly different topographies may still produce the same runoff when these differences are compensated by their associate resistances.

[12]  However, while a physical explanation of the phenomena "landscape organization" is crucial to our understanding, for practical modeling applications we need to step beyond that and actually identify these functional units in the landscape. One avenue is surely to detect these gradients and resistance terms directly based on the available landscape characteristics (*Seibert et al., 2017*). However, it is often difficult to know a-priori which characteristics dominate the function of a landscape element (*Oudin et al., 2010*). Another approach is, hence, to identify functional units a-posteriori directly based on their function, and to subsequently identify which characteristics dominate the hydrological processes, and at which scale (*Sivapalan et al., 2003*). It is exactly here that an information theoretic perspective might be particularly valuable as, despite the more qualitative and descriptive nature of the concept of landscape organization, compressibility is actually quantifiable. For instance two hillslopes showing a similar function with respect to a given process can be compressed and hence combined into a larger landscape element without losing information about the spatial distribution of processes in a catchment. The identification of functional similar areas is hence directly connected to both statistical physics (organization) and information theory (compressibility). For this reason we believe that concepts such as maximum (Shannon)-entropy (*Jaynes, 1957*) and information theoretic variables like the "mutual information" and "Kullback-Leibler divergence" *(Cover and Thomas, 2005; Weijs et al., 2013b; Weijs and van de Giesen, 2013)* provide an excellent framework for connecting the generic informational concepts of statistical inference and compression of data with the specific domain concepts of landscape organization and hydrological similarity.



### 1.4 Objectives and Scope

[13] The main objective of this study is to propose and test a generic approach, based on information theory and to quantifying the relevance and value of spatially distributed data sources for hydrological simulations. Our approach is developed and tested using catchment topography as the source of spatially distributed information, and stream flow and soil water storage as the modeling targets. Specifically, we subdivide a 19.4 km$^2$ catchment into 105 hillslopes and represent each of these contributing spatial units

with a hydrological hillslope model. Following *Loritz et al. (2017)*, the hillslope models are identically parametrized with respect to soils, bedrock topography and vegetation, and differ only with respect to the values of their topography dependent parameters such as aspect, slope and elevation above and distance to the river. Each hillslope model is driven by the same meteorological forcing for one hydrological year, and the Shannon information entropy of the simulation ensemble is analyzed as a function of time.

Further, the similarities of simulated discharge generated by the hillslope models are evaluated in terms of their mutual information, and this is then used as a basis for compressing them into a smaller set of functional groups, such that in each group the members act similarly in terms of runoff generation. Finally, we state that the overall approach presented here is applicable to a variety of different spatially distributed information such as spatially distributed rainfall or land-use, as well as to most modeling target

and to a wide range of spatially distributed hydrological models available. This paper is, however, restricted to development and testing of the approach using only catchment topography and one numerical hillslope model.

## 2. Study area and model realizations

In this section we introduce the study area, the database used, and the general model setup of the different hillslopes.

### 2.1 The Colpach catchment

[14] The 19.4 km$^2$ Colpach catchment is situated in the northern part of the Attert basin in the Devonian schists of the Ardennes massif, and has an elevation ranging from 265 to 512ma.s.l. (Figure 1a).

Approximately 65% of the catchment is forested, mainly on the steep hillslopes. In contrast, the plateaus at the hilltops are predominantly used for agriculture and pasture. The dominant runoff process is rapid flow in a highly permeable saprolite layer above the bedrock, and the catchment is characterized as a fill-and-spill system (*Wrede et al., 2015*). Besides the importance of lateral flow along the bedrock, several irrigation and dye staining experiments have highlighted the role of vertical structures for infiltration and





subsequently for subsurface runoff formation (*Jackisch et al., 2016*). For a more detailed description please see *Loritz et al. (2017), Wrede et al. (2015)* and *Jackisch (2015)*.

## 2.2 The CATFLOW model

[15] The spatially-distributed hillslope-scale model CATFLOW (*Maurer, 1997; Zehe et al., 2001*) is based on the subdivision of a catchment into several hillslopes connected by a drainage network. Each
hillslope is discretized along a 2-dimensional cross section using curvilinear orthogonal coordinates. Each surface model element extends over the width of the hillslope, and these widths may vary along the hillslope. Evapotranspiration is represented using an advanced SVAT approach based on the Penman-Monteith equation, which accounts for tabulated vegetation dynamics, albedo as a function of soil moisture, and the impact of local topography on wind speed and radiation. Soil water dynamics and solute
transport are simulated based on the mixed form of the Darcy-Richards equation, solved using mass conservative Picard iteration and adaptive time stepping (*Celia et al., 1990*). The hillslope module is designed to simulate infiltration excess runoff, saturation excess runoff, re-infiltration of surface runoff, lateral water flow in the subsurface, return flow and solute transport.

## 2.3 Hillslope setup, forcing and model evaluation

[16] The topographic analysis was based on a 5 m Lidar digital elevation model, aggregated and smoothed to 10 m resolution. GRASS GIS (*Neteler et al., 2012*) was used to subdivide the catchment into 105 hillslopes (Figure 1a) and the landscape units mapping program (LUMP; *Francke et al., 2008*) was used to derive the hillslope profiles, including properties such as the elevation and distance to the river, and the mean aspect and width function of each hillslope (Figure 1b). On average the hillslopes lie 67
meters above the river, are 446 meters wide, and cover an area of 0.16 km$^2$. The maximum area of a hillslope is 0.86 km$^2$ while the smallest hillslope covers an area of 0.12 km$^2$.

[17] With respect to soils, bedrock topography and vegetation, the 105 hillslope models were identically parameterized using a parameter set, macropore distribution and subsurface stratification tested and derived by *Loritz et al. (2017)* when representing the entire Colpach catchment by a single effective
hillslope model. Accordingly the hillslopes differ only in the values of parameters that are extracted from the digital elevation model (hillslope profile and length, width and aspect). All hillslope models are 2 m deep, where the upper 1 m is classified as soil followed by a 0.2 m lateral saprolite layer and an 0.8 m deep almost impermeable bedrock (see soil parameter and structure in Tab. 1 in *Loritz et al. 2017*). The porosity of the upper 1 m of soil is assumed to reduce linearly with depth, with the lowest value being 0.3
at a depth of one meter from the surface. In order to account for reported preferential flow in this area *(Jackisch et al., 2017)* we added additionally, every 4 m, a 0.1 m wide rapid flow path (vertical flow structure) with an depth of 1 m. The entire soil setup follows the findings of *Loritz et al. (2017)* in which it



was shown that a representative hillslope was able to provide successful simulations of various hydrological fluxes. The discretization of the hillslope in the downslope direction varies between a maximum of 1 m and minimum of 0.1 m, where the latter occurs close to rapid flow paths. The vertical grid size was set to 0.1 m, with a reduced vertical grid size of the top node of 0.05 m (Figure 1c).

[18]  Boundary conditions were set to an atmospheric boundary at the top, no flow boundary conditions at the upslope, and a gravitational flow boundary condition at the lower boundary. At the hill foot of the hillslope we selected a seepage interface for the upper 0.4 m, where outflow only occurs under saturated and no flow under unsaturated conditions. For the lower 1.6 m of the downslope boundary we selected a no flow boundary to mimic a saturated zone close to the river. All of the hillslopes are covered entirely by forest and the evapotranspiration routine is parameterized similarly to the one described in detail in *Loritz et al. (2017)*. Figure 1c shows an example of a typical CATFLOW hillslope grid and soil setup divided into soil, rapid flow paths and bedrock.

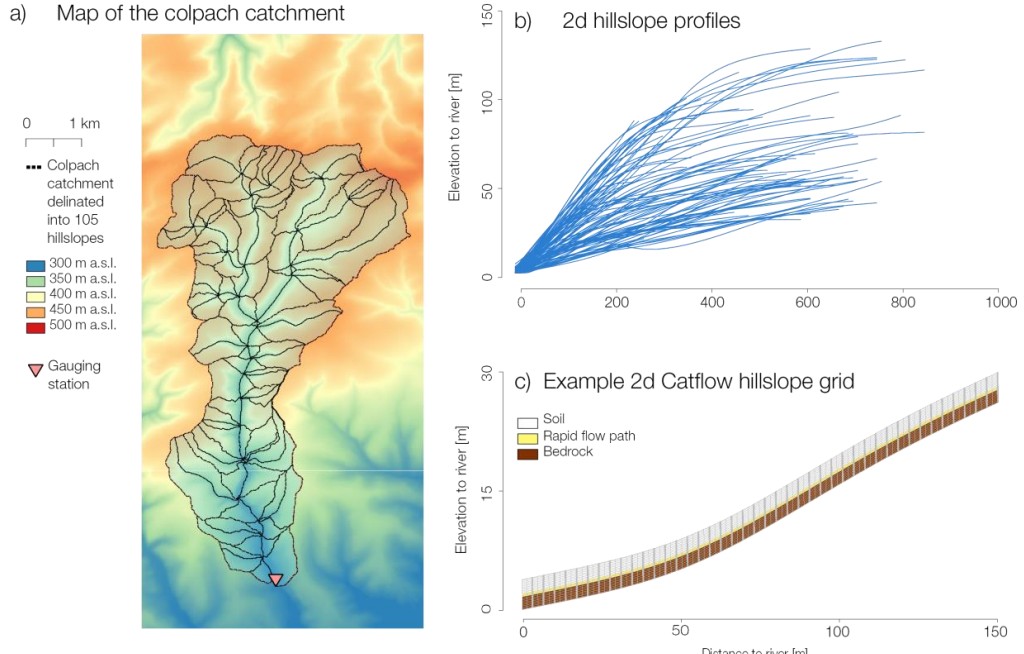

**Figure 1 a) Digital elevation model of the Colpach catchment and its delineation into 105 hillslopes b) all hillslope profiles extracted using the LUMP approach c) example of a CATFLOW hillslope grid.**





### 2.3.1 Model forcing and application


[19] Meteorological input data are recorded at an official meteorological station (Roodt), and were provided by the "Administration des Services Techniques de l'Agriculture Luxembourg". All hillslope models were forced with identical meteorological inputs. This implies, for instance, that we neglect observed variations of rainfall and wind speed within the catchment. We compared simulated and observed specific runoff by dividing the respective values by the relevant contributing areas; i.e., either by the area of the hillslope or of the Colpach catchment. Similarly, we calculated the area specific water storage (average water content per $m^2$) for each hillslope. The simulation period is the hydrological year 2014 from October 2013 to October 2014. This is preceded by a model spin-up of one year with initial states of 70% saturation.


### 2.3.2 Model evaluation


[20] The intention of the model evaluation performed here was not to infer whether we have identified the best performing model structure, but to evaluate and quantify differences in modelled runoff and storage arising from underlying differences in hillslope topography. Therefore, while this exercise does not require a comparison to observations, we nevertheless do so to demonstrate that the different models (and in particular the entire ensemble) produces meaningful simulations that are consistent with observed hydrological storage and streamflow dynamics. We inspected the runoff simulations both visually and by comparison to the observed specific discharge using the normalized mutual information (NMI, specified below; see also *Michaels et al., 1998)*. In addition, we use the Kling–Gupta efficiency (KGE, *Gupta et al., 2009*) to highlight that the NMI provides a consistent picture and is able to identify differences between hydrographs. Furthermore, we use the NMI in our functional classification because it is symmetric and satisfies the mathematical requirements of a distance metric (see section 2.6; for a further comparison of the NMI as well as the Appendix B). Additionally, we calculated the KGE and NMI between the area weighted median of the runoff simulations and the observed specific discharge of the catchment. By simply using the area weighted median instead of a river network routing scheme we assume, in line with *Robinson et al. (1995)* and our own findings *(Loritz et al, 2017)*, that the Colpach catchment is hillslope dominated and that the timing of the routing is small enough to be neglected.




[21] With respect to the storage dynamics, we estimated the average amount of water within the hillslope (in mm for each hillslope) and compared these values against the median of storage estimates calculated from available soil moisture measurements in 10, 30 and 50 cm, which have been collected at different locations throughout the catchment (for detailed information of the soil moisture sensors and observations please see (*Loritz et al. (2017)*). As the model and the observations estimates are based at largely different






scales, we believe that any comparison more detailed than the comparison of their temporal dynamics is in-appropriate.

## 3 Theoretical background, approach and methods

In the following section we provide a detailed review of the important concepts from information theory, and discuss how we used these concepts to address the study objectives.

### 3.1 Information theory and Shannon information entropy

[22] The field of Information theory originally developed within the context of communication engineering, deals with the quantification of information with respect to a concept called "surprise" (*Applebaum, 1996*). For a discrete random variable $X$ that can take on several values $X \in \{x_1; x_2; x_3 \dots x_i\}$ with associated prior probabilities $p(x_1); p(x_2); p(x_3) \dots p(x_i)$ the surprise or information content of receiving/observing a specific value $X = x_i$ is defined as:

$$I = -\log_k\big(p(x_i)\big) \tag{1}$$

where $I$ is the information content, $k$ is the base of the logarithm and $p(x_i)$ the prior probability that $X$ can exist in the state $x$. The logarithm in this definition assures that information is an additive quantity. When the base $k$ of the logarithm is chosen to be 2, information is measured in 'bits' (abbreviated from binary integers). While different k values can be used to calculate the information content of a random discrete variable, here we stick with the logarithm to the base 2.

[23] To calculate the *average* information content associated with the random variable $X$ we can estimate the Shannon entropy $H(X)$ defined (by taking its expectation) as:

$$H(X) = -\sum_{x \in X} p(x_i)\, log_2\, p(x_i) \tag{2}$$

where $p(x_i)$ is again the probability that $X$ can be in the state $x$. In this study we computed the Shannon entropy of the 105 runoff and storage simulations for each hourly time step. In addition to computing the Shannon entropy for a single random variable (also called self-information), we compute the *joint* entropy $H(X,Y)$ of a set of variables $X$ and $Y$ as follows:





$$H(X,Y) = -\sum_{x \in X} \sum_{y \in Y} p(x_i, y_i) \, log_2 \, p(x_i, y_i) \tag{3}$$


where $p(x_i, y_i)$ is the joint probability. The joint entropy is used to estimate the mutual information (described below) between two random variables. For more detailed discussion of information theoretic concepts and variables please see *Applebaum (1996) and Cover and Thomas (2005)*.

### 3.1.1 The appropriate binning for estimating discrete probability density functions

[24] A crucial step in the computation of Shannon entropy and/or mutual information of discrete distribution (see section 3.1 and 3.2) is a careful choice of the bin widths used to construct the probability density functions (pdf; *Gong et al., 2014; Pechlivanidis et al., 2016*). Various guidelines are available regarding how to properly estimate the bin width from the viewpoint of statistical rigor (e.g. *Scott, 1979*). However, *Weijs and van de Giesen, (2013)* also point out that the bin width for a pdf should always be

chosen based on considerations related to the question one wishes to answer. For instance, hydrologists often evaluate their models against measured soil moisture or discharge data. As such observations always imply the existence of measurement errors, observational differences smaller than the typical size of such errors should not be afforded physically meaningful importance.

[25] Accordingly, for calculation of the entropy of the runoff and the storage simulations we propose that

the smallest meaningful bin width should be greater than or equal to the measurement uncertainty. Consequently, we choose the mean relative error of the rating curve (8.5 %, see appendix) to estimate the Shannon entropy of the runoff simulations and the measurement error of the installed capacitive soil moisture probe soil moisture probes of 1 Vol. % for the soil moisture simulations (Decagon 5TE; ± 1 - 2% volumetric water content for calibrated soils; manufacture information). For the runoff simulations, we

started with a bin width of 0.01 mm and then progressively increased the bin width by a factor of 8.5 %. This results in a non-uniform bin width distribution with constantly increasing bin sizes for larger discharge values as the uncertainty in the measurements increases with higher flows. In contrast, for the storage simulations, we used a constant bin width of 10 mm because the measurement errors are constant independently of the magnitude of the measured values.

### 3.1.2 Upper and lower boundary of the Shannon entropy – perfect versus no organization

[26] Isolated systems evolve, according to the second law of thermodynamics, to a state of maximum entropy in which all gradients are depleted and each microstate of the system is equally likely *(Kondepudi and Prigogine, 1998)*. This implies maximum uncertainty about the microstate and the absence of any





organization/order in the system. *Jaynes (1957)* transferred this fundamental insight into a method of

statistical inference, stating *"when making inferences based on incomplete information, the best estimate for the probabilities is the distribution that is consistent with all information, but maximizes uncertainty"*. This condition is reflected by a uniform distribution where all outcomes are equally likely (*Weijs et al., 2010*). With respect to our model ensemble, a state of maximum entropy implies that each of the 105 hillslopes models produces a unique output that cannot be guessed given knowledge regarding the output

of any other hillslope. Accordingly, we can calculate the theoretic maximum entropy for our model study as:

$$H_{max} = log_2 (N) \tag{4}$$

where N = 105 is the number of hillslope models. This maximum reflects a theoretical state of zero spatial

organization in the catchment, where each hillslope provides a unique contribution to stream flow and storage dynamics due to its specific topography. A further compression of the catchment subdivision, for instance by leaving out or merging certain hillslopes, is not possible without losing precision. At the other end of the spectrum, one may have a state of perfect spatial organization in which all 105 hillslope models simulate, within the error margin of observations, identical stream flow and storage dynamics. This would

correspond to zero entropy (the minimum value) and implies that the compression of the spatially distributed model is trivial as any arbitrarily selected hillslope will represent it equally well.

### 3.2 Mutual information as similarity measure

[27] To compare simulated runoff time series generated by different hillslopes, we calculate their pairwise mutual information of each simulated runoff time series as a similarity measure. Mutual information

$I(X, Y)$ between two discrete random variables $X$ and $Y$ is a measure of the strength of their informational correspondence, defined by *Cover and Thomas (2005)* as:

$$I(X,Y) = \sum_{x \in X} \sum_{y \in Y} p(x,y) \, log_2 \frac{p(x,y)}{p(x) \, p(y)} \tag{5}$$

where $p(x,y)$ is the joint probability of $X$ and $Y$ and $p(x)$ and $p(y)$ are their marginal probabilities.

Equivalently, mutual information can also be calculated directly as a difference between the sum of the entropies of $X$ and $Y$ minus the joint entropy of $X$ and $Y$ (Figure 2).





$$I(X,Y) = \ H(X) + H(Y) - \ H(X,Y) \qquad (6)$$

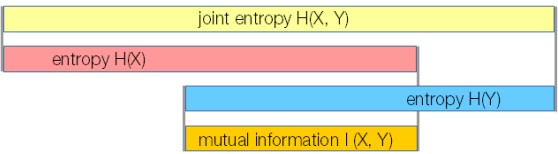

**Figure 2 Sketch of the relation between information entropy, joint entropy and mutual information displayed as bar diagram.**

[28]  While Shannon entropy is used to determine the information redundancy or compressibility between the 105 simulated discharge time series at a certain time steps, we now show how mutual information can be used to see how similar or dissimilar two discharge simulations are.

[29]  Mutual information quantifies the amount of information that one variable reveals about another and thus the strength of their co-dependence. If the mutual information is zero, the two variables are independent while larger values correspond to stronger relationships. When using the binary logarithm mutual information, Shannon entropy and joint entropy share the same unit 'bits'. Here, we seek to use the mutual information between different hillslope runoff simulations as a measure of similarity or distance between the hillslope models. However, since the value of mutual information depends on the absolute magnitude of joint entropy between the two chosen variables, it is not appropriate to use mutual information directly as a distance function for relative comparisons (*Michaels et al. 1998*); e.g., if the joint entropy of two variables is low the value of mutual information will also be low even if the two variables are perfectly related. Hence, following *Michaels et al. (1998)*, we normalize $I(X,Y)$ using the larger of the entropies of the two random variables *X* and *Y*.

$$NMI(\text{X}, \text{Y}) = \frac{\text{I}(\text{X}, \text{Y})}{\max[\text{H}(\text{X}), \text{H}(\text{Y})]} \qquad (7)$$

[30]  Accordingly, the normalized mutual information (NMI) ranges from 0 to 1, with higher values corresponding to stronger relationships (higher mutual information content). With this normalization, the





NMI becomes symmetric and satisfies the mathematical axioms of a distance function. Further, since distance functions are intuitively expected to be closer to zero in case of a stronger similarity, we subtract the NMI from 1 (to get 1-NMI (see Appendix B for a comparison of the NMI with the Pearson correlation coefficient and the Euclidean distance)).

### 3.3 Functional classification of hillslopes with similar runoff behavior

[31] Using NMI as distance metric, we classified the 105 hillslope models into functional groups of similar runoff behavior based on the 105 runoff time series, using a hierarchical cluster analysis based on Ward's minimum variance method (*Hastie et al., 2009; Murtagh and Legendre, 2014*). As a first guess of a physically meaningful number of functional groups we used the mean annual entropy of all 105 discharge simulations (further discussed in section 4.2).

$$No. of\ functional\ groups = 2^{(mean\ annual\ entropy)} \tag{8}$$

[32] This choice is inspired by the fact that the Shannon entropy of a random variable $X$ is closely related to the maximum compressibility of the information about this variable. This is because, when the Shannon entropy is calculated using the binary logarithm, it relates to the minimum number of binary "yes or no questions" necessary to determine the actual value of $x_i$ from $X$. In the special case where the distribution of the random variable is dyadic, the value of the Shannon entropy $H(X)$ and the expected minimum number of questions are equivalent, while if this is not the case the expected number of questions lies between the computed value of the entropy H and its increment H+1 (for further details see *Cover and Thomas, 2005*).

$$H(X) \leq\ Expected\ Questions\ <\ H(X) + 1 \tag{9}$$

[33] So, in general, if the entropy of a discrete random variable $X$ is $H(X) = 2$, we know that the expected number of binary (Yes/No) questions needed to quantify x lies between 2 and 3. This implies that the number of possible outcomes lies somewhere between $2^2 = 4$ and $2^3 = 8$, as every binary question can have two possible answers.



### 3.4 Compression of the catchment model based on functional groups

[34]  Having grouped the hillslope models into time-invariant functionally similar groups, we test whether this grouping provides a solid basis to compress the model structure of 105 hillslopes into a less redundant one that yet produces results of similar quality as the full set of hillslopes but at much smaller computational cost. There are at least three avenues to do so. The first one is to simply calculate the area weighted median or average of all runoff simulations within a functional group. This, however, means that all 105 runoff simulations are necessary to build this compressed model and we cannot run the compressed model in a forward mode. The second avenue is to take functionally united hillslopes and derive for each functional unit an effective, spatially aggregated hillslope in a similar fashion as done in *Loritz et al. 2017*. Though this is most likely the most promising way to come up with a compressed catchment model, it is beyond the scope of this manuscript. Instead, to simplify this attempt we use a third option and develop a compressed model structure using a bootstrap-like approach. For this we randomly select a single hillslope from each functional group, and calculate the area weighted median of the simulated discharge time series of the six randomly selected hillslope models (Compressed catchment model; Figure 3). The weight assigned to each of the selected discharge time series corresponds to the areal fraction of all hillslopes in the respective functional group. This assures mass conservation because runoff of each hillslope is equal to its area times the simulated specific discharge. We use random selection because each group member is regarded as equivalent to represent the runoff generation of the corresponding functional group. To account for sampling variability, as simulated runoff differs slightly among the hillslopes within a functional group, we repeat this random selection 1000 times. In a final step, we compare those values individually as well as the median of all realizations against the observed runoff of the Colpach using the KGE. This reveals the performance spread of the randomly generated compressed models compare to the area-weighted median of the entire 105 hillslopes.

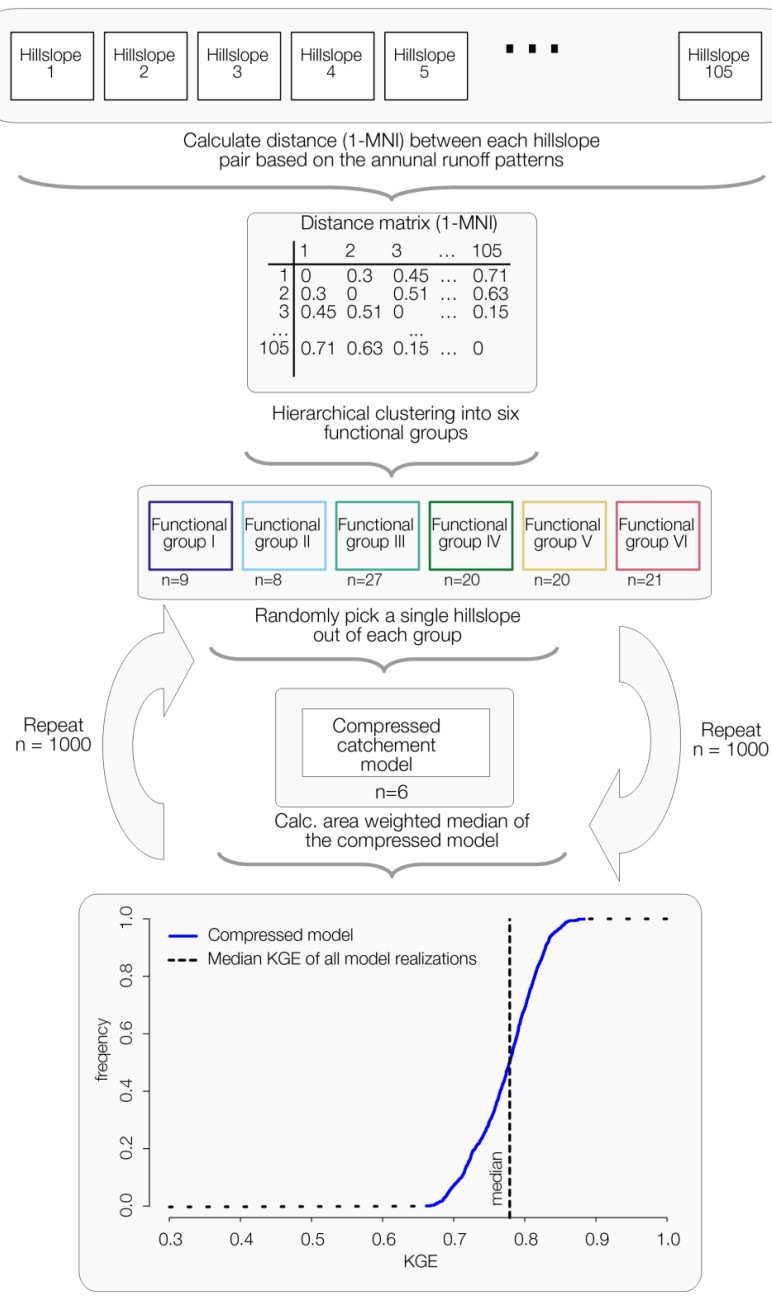

**Figure 3 Sketch of the approach for compression and performance evaluation for the compressed catchment model.**





## 4. Results

### 4.1 Runoff and storage simulations


[35] The ability of different hillslope models to reproduce the observed runoff dynamics of the Colpach catchment varies substantially (see Figure 4a), with KGE values ranging between 0.44 and 0.92. This apparent spread in model performance among the hillslopes corroborates the sensitivity of simulated discharge to those parameters derived from the DEM. A similar pattern is revealed when model

"goodness" is expressed by means of the normalized mutual information (NMI) between each hillslope model and the observed runoff. NMI values range from 0.51 to 0.71 and show a strong linear correlation to the corresponding KGE values (with a Pearson correlation coefficient of 0.89). This good correspondence of NMI with the KGE performance measure reinforces the notion that NMI is a suitable measure of similarity, or difference, between time series of hydrological variables.

[36] The temporal patterns of total area specific storage for each hillslope model are shown in Figure 4b. The skill of different hillslopes to reproduce the temporal dynamics of observed median storage is rather stable, with a Spearman rank correlation coefficient ranging from 0.77 to 0.86, with the ensemble median having a value of 0.82. Visual comparison of the simulated storage time series reveals that differences in hillslope topography result mainly in a parallel shift of the respective time series. This parallel spreading is

stronger during the wet season and less pronounced during dry conditions. The latter might be due to the identical vegetation parameterization of each hillslope and hence a result of highly similar root water uptake which dominates storage dynamics during dry conditions in summer.





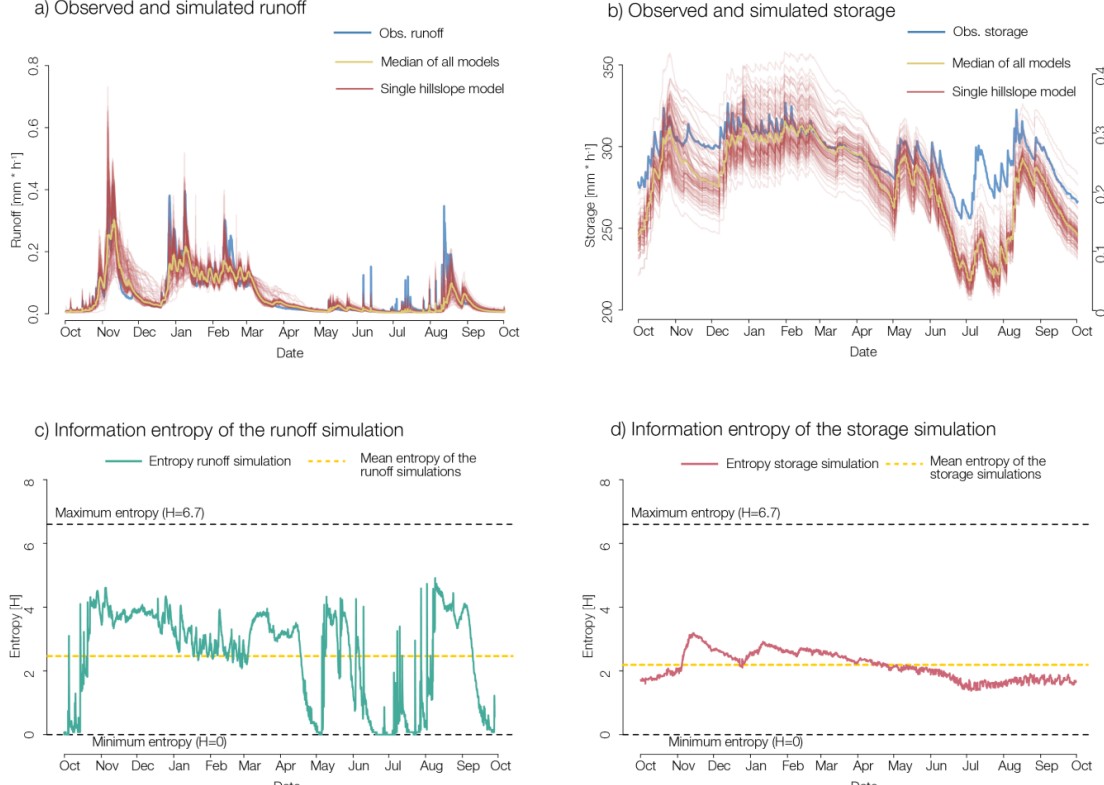

**Figure 4 (a) Observed and simulated runoff of the Colpach catchment. The red lines correspond to individual hillslope models and the yellow line to area weighted median of all hillslopes. (b) Simulated total area specific storage of each hillslope in red and the median of all models in yellow. The median of the 141 observed soil moisture time series is smoothed with a 12 hour rolling mean (for more detail to the soil moisture observation we refer to (*Loritz et al., 2017*)) (c) Shannon entropy in turquoise for the runoff simulations as well as the corresponding mean and (d) a similar plot for the storage simulations (red).**

## 4.2 Entropy of the discharge and storage simulations

[37] If all 105 of the hillslope models were to produce unique simulations of equal importance, their entropy would be the theoretical maximum value of 6.71. Similarly, the minimum value associated with a perfectly redundant set of hillslopes, is 0. As seen in Figure 4c & d, the entropy of the ensemble of runoff simulations starts at a rather low value at the beginning of our simulation period, increases with the first rainfall events in autumn, stays at a high level (ranging between 3 and 4) during the winter period, and starts to decrease towards 0 in May. During the summer, the entropy reacts much more strongly to the



different rainfall events than in winter, and peaks at a value of 4.9 in August when stream flow production

grows again after a long dry period of low flow. It is interesting to note that the entropy in simulated stream flow is highly dynamic in time, implying that the required structural resolution of the model changes with time, with the 105-hillslope model structure being less redundant during periods of high entropy and more strongly redundant when entropy approaches 0 (see also Appendix C).

[38]   For the ensemble of storage simulations, the entropy varies between 1.5 and 2.9, which indicates less

temporal variability compared to runoff. This is consistent with the visual impression that differences in topography result mainly in a parallel shift of the time series to a different annual mean. Nonetheless, the entropy time series exhibits weak annual dynamics, with a peak in mid-November when the wet season starts. This peak coincides with the entropy peak of the runoff simulations. In spring and summer, the entropy decreases slowly until it reaches the overall minimum of 1.71 in October. Note that this could be

very different in case of (for instance) land-use differences or distributed rainfall among the hillslopes causing a likely increase of entropy during summer and autumn.

### 4.3 Functional group and their typical runoff and storage dynamics

[39]   The mean annual entropy of the runoff simulations is 2.5 (Figure 4c), which implies that (on average) the number of functional groups or bins that can be distinguished lies between $2^{2.5} \approx 6$ and $2^{3.5} \approx 10$.

In line with one of our goals to use information theoretic measures to define similar acting landscape elements and to compress the full catchment model into functional groups without substantial loss of information we took the lower value and used a hierarchical cluster analysis to classify the hillslopes into six functional groups using normalized mutual information efficiency (1-NMI) as distance metric. The median discharge for each functional group is shown in Figure 5a, while the corresponding set of hillslope

profiles is displayed in Figure 5b. In general it seems that the functional groups 1, 2 and 6 exhibit the strongest differences with respect to their median runoff time series as well as with respect to the geopotential profiles whereas the classes 3, 4 and 5 appear much more similar in both aspects. The median of the storage simulation of each functional group is displayed in Figure 5c. Consistently with simulated runoff, the storage time series of functional groups 1, 2 and 6 show the greatest differences. However, in

contrast to the runoff simulations also the functional groups 3, 4 and 5 are better separable at least during the wet period. Consistent with the decline of the Shannon entropy in Figure 4d these differences diminish in summer. Especially in June, July and August all of the functional groups simulate essentially identical storages as their differences are getting closer to the error margins of the soil moisture measurements. Again, we stress that this convergence could be explained by the dominant role of evapotranspiration and

the identical land-use parameterization of all hillslopes. Note that functional group 6, showing the strongest and fastest overall runoff reaction and has the lowest overall storage simulation. Consistently




with that, functional group 1 and 2, showing the slowest runoff reaction are characterized by the highest overall storage.

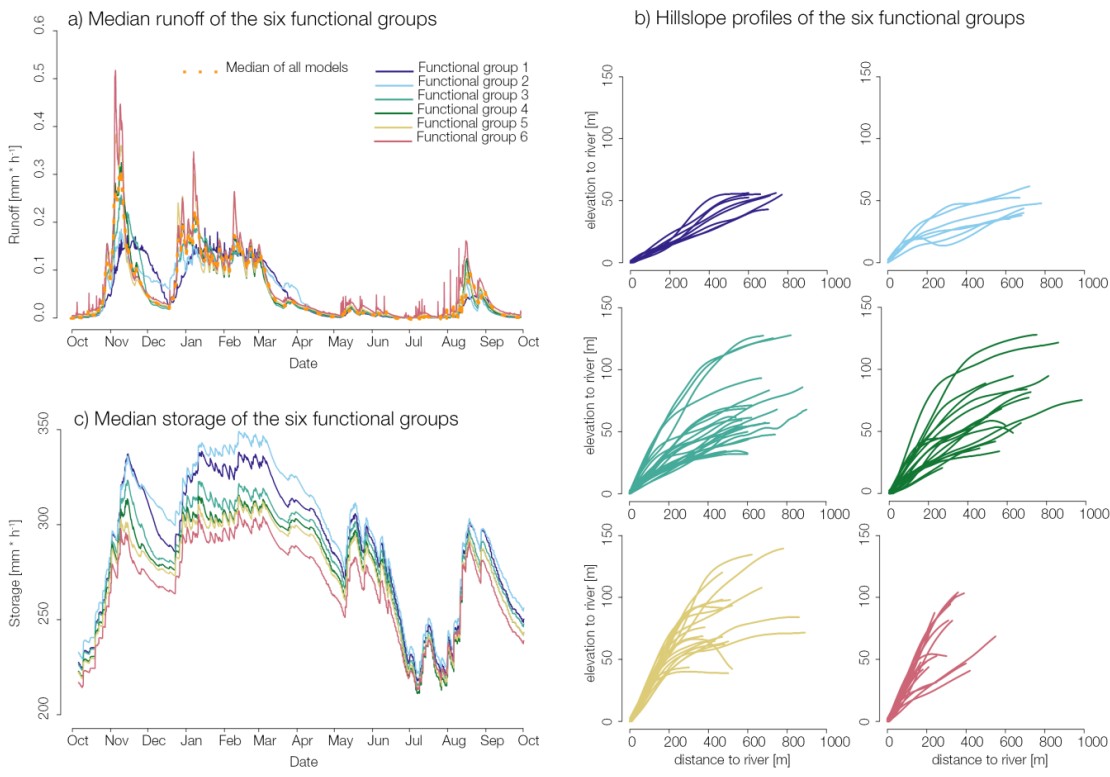

**Figure 5 a) Median runoff of the six functional groups; b1 –b6) corresponding hillslope profiles with the elevation to river on the y axis and distance to river on the x axis for each functional group. c) Median storage of the six functional groups.**

## 4.4 Performance of the compressed catchment models

[40]  Figure 6 shows the cumulative frequency distribution of KGE values for the 1000 randomly selected model compressions using the aforementioned functional groups of similar runoff generation (Table 1). The median KGE of all trials of 0.78 corroborates that the compressed model structures perform on average slightly better than the area weighted median of the 105 hillslope models, which has a KGE of 0.76. However, the range of 0.66 to 0.88 in the KGE values indicates that the performance of a particular single realization of the compression depends on the actual combination of hillslopes selected for each



group. As each realization of the compressed catchment model would in principle only use six hillslope models and if we assume that all hillslopes have the same run time this could, in theory, reduce the computational costs of our model application by a factor of 17.5. The latter means a reduction in computational cost by two orders of magnitude.


**Table 1 Number of member as well as the mean and max values of the runoff simulation of each functional group.**

| Functional group | Group 1 | Group 2 | Group 3 | Group 4 | Group 5 | Group 6 |
|---|---|---|---|---|---|---|
| member [n] | 9 | 8 | 27 | 20 | 20 | 21 |
| mean annual runoff [mm*h$^{-1}$] | 0.051 | 0.052 | 0.053 | 0.054 | 0.056 | 0.065 |
| max runoff [mm*h$^{-1}$] | 0.22 | 0.34 | 0.42 | 0.43 | 0.64 | 0.75 |
| mean storage [mm * h$^{-1}$] | 289.6 | 295.7 | 281.7 | 277.1 | 273.7 | 267.7 |
| max storage [mm*h$^{-1}$] | 338.6 | 349.1 | 323.7 | 316.2 | 312.8 | 307.2 |

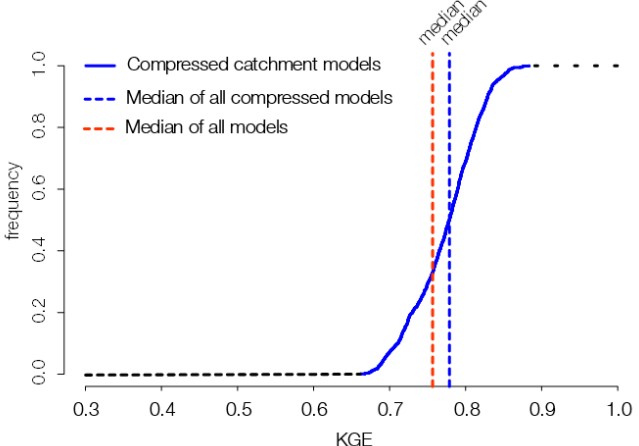

**Figure 6 Distribution of model performances of the different realizations of the compressed catchment model (blue). The tow dashed lines correspond to the median of all realization of the compressed catchment model (blue) as well as to the area weighted median of all 105 hillslope models (red).**




## 5. Discussion

[41] The results presented above provide strong evidence that information theoretic concepts are powerful tools to quantify and explain the relevance of different system characteristics for distributed modelling. Following this overall result, we will start to discuss our main finding that the amount of topographic information relevant for distributed modelling is not constant but time variant. Furthermore in a second step, we address the closely related issue that we are able to compress the ensemble of hillslope models into functionally similar groups, and that a stronger compressibility implies a higher degree of functional organization in a heterogeneous environment. This discussion leads naturally to a short reflection on the advantages that concepts from information theory offer for exploring and explaining how spatial complexity and functional similarity of hydrological systems are connected. Finally, we conclude by revisiting the seeming antagonism between landscape organization (*Dooge, 1986*) and functional similarity (*Wagener et al. 2007*) against the recurring finding of heterogeneity and randomness and hence uniqueness of hydrological places (*Beven, 2000*) and provide an outlook on how to generalize the approach presented here.

### 5.1 Temporarily varying importance of topography for distributed modeling

[42] The relevance of spatially variable but yet time-invariant topographic information on hydrological simulations was found to be strongly time dependent. The different topographic information used within the models led to complex temporal dynamics of the information content within the hillslope responses and those were distinctly different for the two target variables. The Shannon entropy of the discharge simulations revealed that there are alternating periods of high similarity and of high diversity among the hillslope responses. This resulted in several local maxima and absolute minima. These maxima and minima are not easily explained by simply attributing them to high and low flow conditions (see Appendix C). For example the global maximum of 4.9 (close to the theoretical maximum of 6.7) was observed in August, when the system rapidly switched from low to high streamflow conditions in response to a strong convective rainfall event. In contrast, the Shannon entropy of storage simulation exhibited a distinctly different pattern compared to the discharge simulations with a much stronger autocorrelation, two clear identifiable maxima in winter, and overall lower values of the Shannon entropy in summer.

[43] The overall differences between the two target variables, the dynamics of the information content within the discharge and storage simulations, and hence the changing maximal compressibility of the model ensemble, highlights that the relevant topographic information for distributed modeling depends firstly on the modeling target and secondly on the time, and thus on the prevailing forcing as well as on the state of the system. In other words, spatially distributed information about topography has a time





varying impact on the model ensemble. Hence, the necessary complexity (*Schoups et al., 2008*) of a distributed model to capture this information is time dependent as well.

[44]  If we try to generalize and transfer this finding from the model world to a real hydrological system keeping in mind all the issues that go along such an approach, these results imply that *different landscape entities may either function similarly or dissimilarly depending on the time*. Hydrological similarity can

therefore, rather than being static, be a dynamic attribute that depends on the "*hydrological context*". Interestingly, this context dependence can be straightforwardly explained by the generally dissipative nature of hydrological processes (*Kleidon, 2010*). Rainfall and radiation push and pull the hillslopes away from their local thermodynamic equilibrium, thereby generating internal system gradients in either potential energy or capillary binding energy. These gradients get depleted during system relaxation

towards the equilibrium either through release of water from hillslopes to the stream or through recharge and capillary rise (*Zehe et al., 2014*). However, the generation and depletion of these gradients is controlled by a large variety of meteorological and hydrological processes interacting across a hierarchy of spatial and temporal scales (*Blöschl and Sivapalan, 1995*). Exactly the varying dominance of these processes, and hence the changing importance of the corresponding landscape control, is the key to

understanding the time varying relevance of different system characteristics for distributed hydrological modeling, and explains the varying relevance of (in our case) topography for hydrological modeling even though topography is quasi static at classical hydrological time scale.

### 5.2 Compressibility of time series and functional similarity of hillslopes

[45]  As indicated in the section above, both of the target variables, storage and discharge, never reached

the theoretical maximum value of the Shannon entropy implying that the model ensemble was producing redundancy and thus was compressible during the entire year. Based on this general finding we came up with the idea of a compressed catchment model which was built upon a straightforward clustering of all hillslope models into functional groups of similar annual runoff behavior. This compressed model consisted in a single realization of 6 instead of 105 hillslopes, which were then randomly drawn from each

functional group. It is of interest that by reducing the model ensemble to a smaller set of hillslope models we were still able to match on average the observed annual streamflow in the catchment. This result agrees with the findings of *Fenicia et al. (2016)* who stated that spatial variations of the geopotential are too small in this landscape to have a dominant influence on the annual runoff generation, and with the findings of a foregoing study where we show that the annual runoff dynamics of the Colpach catchment

can be simulated using a single effectively compressed hillslope model (*Loritz et al. 2017*).

[46]  Neglecting all the issues that occur when we compare distributed model applications with spatially aggregated models (*e.g. Obled et al., 1994; Beven and Freer, 2001; Pokhrel et al., 2012*) our comparison



of the differently strong compressed catchment models matches with the conclusion of *Pokhrel and Gupta (2010)* that as long as we are not interested in the representation of the spatial distribution of hydrological

fluxes or state variables, a spatially aggregated model which compresses the spatial variability of the landscape properties might be sufficient for predicting macroscopic variables (*Hrachowitz and Clark, 2017*). However, as soon as our focus shifts to the representation of the spatial distribution of a hydrological process, information entropy bears the key to defining and diagnosing the minimum adequate complexity of a distributed model (*Schoups et al., 2008*), particularly as it could help guide an approach to

reducing computational costs without losing information (in our case by a factor of almost 17.5).

[47] However, the assessment of a meaningful compression that leads to a less redundant and yet well performing distributed model structure is not at all a straightforward exercise. This is corroborated by the strongly variable performance of the 1000 randomly generated compressions, which highlights that the individual performance depends strongly on the model realization. From this we conclude that, contrary to

our assumption, not each hillslope model represents stream flow generation of a functional unit equally well, as our classification is based on mutual information between the annual discharge time series. The fact that two hillslope models may yet act differently at certain time steps explains why every random realization of the model compression performs slightly different. The second and maybe more general shortcoming is that our proposed compression is based on a fixed number of groups, inferred from the

average annual entropy. As the average annual entropy of simulated streamflow reflects the annual average maximal compressibility of the discharge simulation, our choice for the number of functional groups seems legitimate as a first attempt on an annual scale. However, as shown in Figure 4c the Shannon entropy of the discharge simulations deviates substantially from this value. This implies that our model structure is either too simple in periods where the entropy is larger than the average or redundant in

periods where the entropy is smaller. A best possible compression of a distributed catchment model, defined as the one that avoids any loss of information and also avoids any redundancy (also referred as lossless compression e.g. *Weijs et al. (2013b)*) will therefore require a time variant number of functional groups. Such an effort to do simulations with a higher spatial model resolution in times of high spatial organization and with a coarser spatial model resolution in times of low spatial complexity, as is for

example done with different adaptive time stepping schemes in numerical model implementations (e.g. *Clark and Kavetski, 2010*), points to new challenges that are not only beyond the scope of this study but likely also beyond the capabilities of most currently available model systems.

### 5.3 Information theoretic measures to quantify similarity

[48] The venture to link complexity of spatially distributed catchment characteristics to functional

similarity led us naturally to the concepts of information and (physical) entropy *(Davies, 1990; Ben-Naim, 2008)*. Similarity of runoff, or storage of hillslopes, implies that their contribution to streamflow is





redundant and hence does not change the information entropy within the simulations beyond its areal share (at least as long as the timing of the routing is not dominant). Removing this redundancy means to compress *(Weijs et al., 2013a)*, and in our specific case to aggregate hillslopes to larger similar functioning landscape elements which we called functional groups in relation to the definition of functional units by *Zehe et al., (2014)*. Although it is evident that this partitioning of similar acting units into larger groups does not require the use of information theory (e.g. *Wood et al., 1988, Sawicz et al. 2011,* Berghuijs et al., 2014), we believe that, besides the maybe more general assets of an information theoretic perspective on different hydrological issues (e.g. *Weijs and van de Giesen, 2013*, *Gupta and Nearing, 2014; Ehret et al., 2014; Nearing et al., 2016*), it has also major technical advantages for a variety of different tasks as shortly discussed in the following.

[49]  First, information theoretic measures like Shannon entropy and mutual information, when calculated with the same logarithmic base, share the same units, in our case "bits". This facilitates the inter-comparison of the different variables, in our case storage and runoff, with respect to their diversity in the model ensemble without the necessity of normalization. Furthermore, if calculated in the discrete form, a careful choice of the bin width according to the measurement error can also be interpreted as physical meaningful definition of the minimum separable difference between observations or simulations of the same state variable or flux. For instance, in this study, we used the inherent measurement errors of the soil moisture probes as well as the uncertainty in our rating curves to define the minimum separable differences of storage and runoff.

[50]  Another key advantage of the information theoretic perspective is that not only the minimum but also maximum information content and hence the maximal complexity or functional disorganization that a distributed model can produce in its responses is well defined. The latter corresponds to the state of maximum Shannon entropy which implies that each time series, either modelled or observed, contributes in a unique (non-redundant) fashion to the ensemble. We are therefore able to derive a theoretical upper and lower bound which reflects naturally the minimum and maximum reachable complexity of state/output response that our model can produce. The lower boundary represented by a zero entropy, corresponds to a situation where all model elements produce with respect to the corresponding observation error the same output and hence act identically. The upper boundary or maximum entropy, in our case 6.7, corresponds to a situation where all model units produce a unique output and to a situation of no redundancy at all. Given these two margins we can judge whether different model elements, in our case hillslopes, of a chosen model provide largely independent stream flow contributions or the other way around.



## 6. Conclusion and Outlook

[51] Based on the evidence presented here, we conclude that the proposed information theoretic measures and concepts provide a powerful framework for understanding and diagnosing how landscape organization and functional similarity of hydrological systems are connected. We are aware that the specific findings of the present work are necessarily constrained by the a-priori settings of the model ensemble, which exclusively focused on a spatially variable topography, while land-use, precipitation and

the soil parameters were identical among the 105 hillslopes. The application of these concepts and the general mindset is, however, by no means restricted to this specific model neither to topography. On the contrary, it may be generalized either by additional data sources such as land-use, bedrock topography and distributed rainfall data as well as to any ensemble of time series, modeled or observed. This opens new opportunities to systematically explore how spatial variations of different landscape characteristics and

meteorological forcing affect hydrological processes. Furthermore, as we only tested first order changes of topography and the influence on distributed modeling here, it also opens the possibility to test whether second order effects arise from combinations of several distributed characteristics.

[52] Finally, in line with *Clark et al.(2016)* we argue that a comprehensive answer to the simple question stated in the introduction "*when is the spatial variation of a system characteristic large enough that we*

*need to account for it*" is not at all straightforward, but requires a solid theoretical framework. Following thermodynamic reasoning and information theory, the key to explain why hydrological systems often act so comprehensibly is that they are dissipative and highly organized *(Zehe et al., 2014)*. This implies that organized simplicity might emerge when we move up to larger scales in space *(Dooge, 1986; Savenije and Hrachowitz, 2017)*. Our results reveal, however, that simplicity manifests not only in space when moving

to larger scales, but also manifests when "the system moves through time" as functional similarity emerges in time. We therefore propose that the concepts of landscape areas that act either similarly and are thus redundant *(Wagener et al., 2007)* or show unique functioning and are thus irreplaceable (*Beven, 2000*) are consequently not mutually exclusive. They are in fact of complementary nature, and systems operate by gradually changing to different levels of organization in which their behaviors are partly unique

and partly similar.





*Data availability.* The hydrological model 'CATFLOW' as well as all simulation results are available from the leading author on request. For the soil moisture observations please contact Markus Weiler (University of Freiburg) or Therea Blume (GFZ Potsdam) and for the discharge observations please contact Laurent Pfister or Jean-Francois Iffly from the Luxembourg Institute of Science and Technology.


# Appendix

## Appendix A: Uncertainty of the rating curve

[53]  For the gauge "Colpach" the rating curve was given with:


$$(11) \qquad\qquad Q = 10.59 * (h - 0.11)^{2.14}$$

where Q is discharge ($m^3*s^{-1}$) and h is gauge level (m). It was derived by ordinary least square fitting to 15 direct discharge measurements (Figure. 7 green dots). Using the rating curve for flood frequency analyses would require a validation against an independent set of direct discharge measurements (grey dots). In

order to we use it as proxy for the binning width to estimate the pdfs, we calculated its overall uncertainty relative to the total set of direct discharge measurements (green and grey dots) as RMSE with a value of 8.5% (dashed red line).

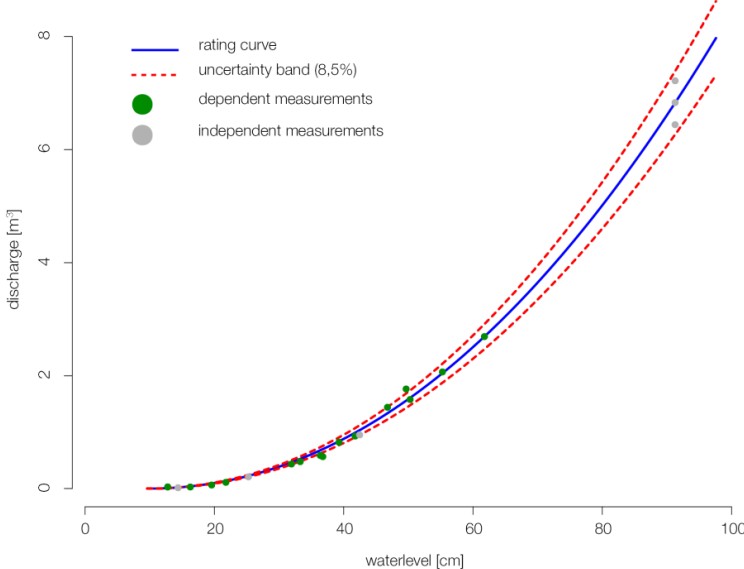

**Figure 7 Rating curve of the Colpach gauge. Green dots which were used to estimate the rating curve, gray dots**
**independent discharge measurements.**



**Appendix B: Comparison of the NMI**

[54] To illustrate the performance of this metric, Figure 8 shows a comparison of normalized mutual information (NMI) to the Pearson correlation and the Euclidean distance for four different synthetic cases:

(a) linear relationship between $X$ and $Y$

          (b) difference between two sinusoidal functions with different amplitudes

          (c) quadratic relationship between $X$ and $Y$

          (d) two independent random variables $X$ and $Y$

[55] We used equally distant bin widths of 0.05 to estimate the pdf for the calculation of the mutual

information in all four cases.

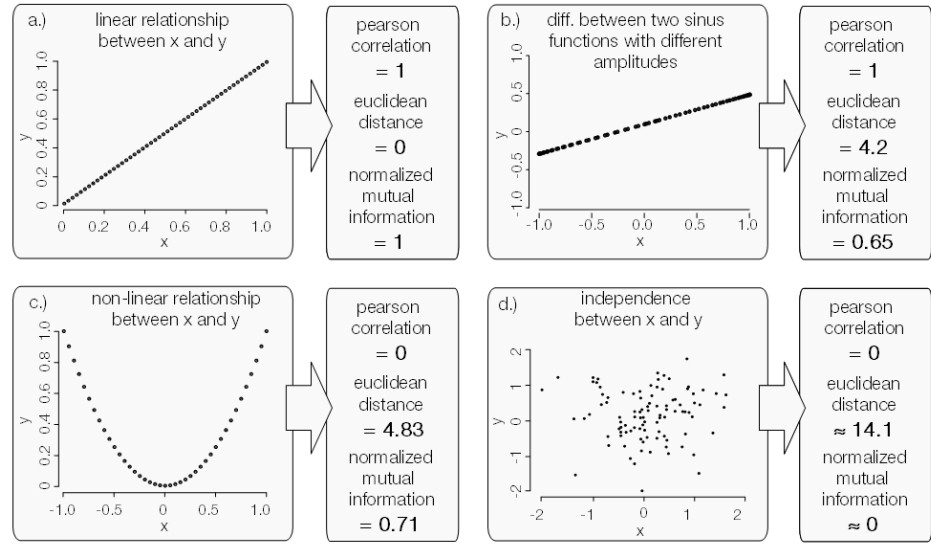

**Figure 8 Difference between the Pearson correlation coefficient, Euclidean distance and the normalized mutual information. Four cases are shown (a) linear relationship, (b) the difference between two sinus functions with different amplitude, (c) a quadratic relationship and (d) two independent variables. The pdf was estimated using an equally distant**

**bin width of 0.05 in all four cases.**

**Appendix C: Shannon entropy of the runoff simulations against the median discharge of the runoff simulations**

Relation between the area-weighted median of the discharge simulation against the Shannon entropy of all

discharge simulations for each time step (Figure 9). The graph highlights that there is no simple linear relation between discharge height, time of the year and the Shannon entropy.





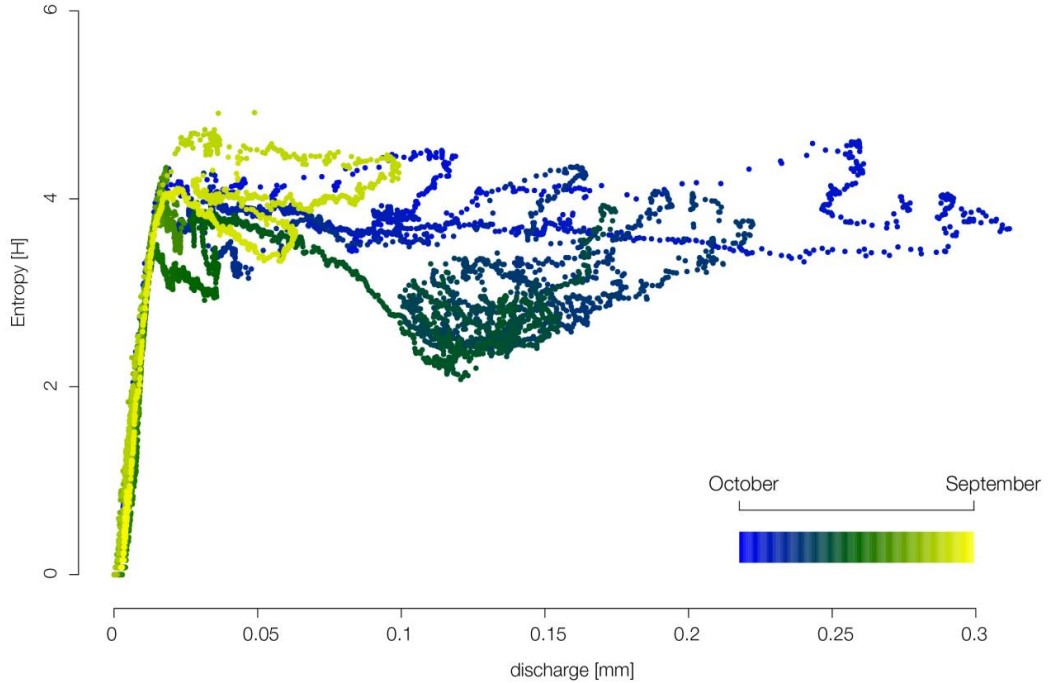

**Figure 9 Shannon entropy of the 105 discharge simulations against the area-weighted median of the discharge simulations. The color key range from blue (winter) over green (autumn / spring) to yellow (summer) and illustrates the time of the year.**




*Acknowledgements.* This research contributes to the "Catchments As Organized Systems (CAOS)" research group (FOR 1598) funded by the German Science Foundation (DFG). Laurent Pfister and Jean-Francois Iffly from the Luxembourg Institute of Science and Technology (LIST) are acknowledged for organizing the permissions for the experiments and providing discharge
data for the Colpach. We also thank the whole CAOS team of phase I & II. Particular we thank Malte Neuper (KIT) for support and discussions on the rainfall data and Markus Weiler (University of Freiburg), Theresa Blume (GFZ Potsdam) and Britta Kattenstroth (University of Freiburg) for providing and collecting the soil moisture data.

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
