# Peer review of "On the dynamic nature of hydrological similarity"

_Hydrology and Earth System Sciences, 2017_

## Referee Comment (RC1) · S.V. Weijs (Referee) · 27 Mar 2018

The paper presents an interesting experiment in evaluating the needed spatial complexity of hydrological models, using an information-theoretical approach to compare a "full" model with a compressed one, and testing for similarity in responses to define simpler models. Generally, the paper is well-written, with interesting thoughts put forward in the discussion. I found the idea very relevant, and think there is a large potential for further work in this direction, perhaps focusing also on predictability rather than mostly on similarity. For the current paper, I have a number of comments/questions that I think would need some clarification in the paper. The comments are itemized below. I think most of the suggestions could easily be addressed/clarified in this paper, while some points may be more suitable to be addressed in further work.

1) On line 75 it is mentioned that some of the energy is dissipated in runoff concentration, and the remainder is exported as kinetic energy of streamflow. I would say that the vast majority of the energy is dissipated and only a tiny remaining fraction exported. If I am not mistaken, a back of the envelope calculation with reasonable (conservative) numbers : average height of catchment above river = 100 m, 100% runoff ratio, and 2 m/s flow velocity, gives me that 99.8% of energy is not exported as kinetic energy by stream flow. To dissipate a smaller portion, we would have to build large constructions (dams, penstocks) to get more kinetic energy and utilize for hydropower, and that is usually mainly saving the in stream dissipation, not much of the dissipation from runoff concentration.

2) 154: I would say that in hydrology, next to resistances, there is a large influence of storages (or capacitors in an electronics analogy) in the system that would make the dynamic behaviour unique/ different between two cases with the same driver/resistance combination. Could it be that the capacitances have some relation with the resistances (e.g. through porosity and path length through soil), that explains part of the similarity?

3) 222: Could you briefly state what criteria were used for the subdivision in hillslopes? I would say that could be quite relevant to understand possible redundancy/similarity. Is the stream network based on an algorithm or on direct observation in the field? (see e.g. Mutzner et al, 2016)

4) 328: Apart from the precision of the data, another very important consideration in the number of bins chosen is how the number of bins compares to the number of data points (this is the same consideration that goes into choice of bins when presenting data in a histogram). If bins are too fine for a relatively low number of data points, this will result in values always close to log(m), where m is the number of data points. If too coarse, we lose information that was in the distribution. It would be helpful to mention the number of bins used for both variables, and the number of data points, to clarify that the two are in balance. Perhaps also some discussion is warranted about using constant bins over the year vs. rescaling bins every time step using the maximum model. As noted, this choice basically reflects the question of which we quantify the

information contained in the answers to it.

5) 334: It is not completely clear to me if/how the soil moisture probe uncertainty is used in the calculation of the bin width. Is it a relevant factor in the bin width if observations are not directly used to compare with the simulations?

6) 352: The upper bound is not only related to the number of models, but also to the number of bins used to discretize the values. Now, if I understand well, the number of bins is unbounded, but in practice, there is only a maximum number of bins occupied (would be good to know how many). Depending on the range of the bins, in order to reach the maximum entropy, the models may need to simulate unrealistically high or low values. Therefore I have some problem with the interpretation of log(N) as the maximum attainable entropy / state of zero spatial organisation.

7) 348-350: I think there is also another key issue here: predictability from other hill-slopes is not the same as giving identical unit runoff values. If e.g. runoff follows the same pattern with a simple scaling factor, a time shift, or any one-to-one relation, I would say the hillslopes are redundant to some extent. Giving identical runoff values is a sufficient, but not a necessary condition for providing information that is redundant.

8) 356: but attaining this maximum would also mean the rain has to be white noise spatially.

9) 357: precision: I am not sure if this statement is accurate, see point 7.

10) 359: identical : I think spatial organization/compressibility is more related to how predictable one hillslope output is from the other, rather than how identical they are, see also point 7.

11) 442: The median of all realizations is not a compressed model, since it would need calculation results from all catchments. Please clarify whether you calculated: a) the KGE of the median of the simulation outputs from the 1000 bootstrapped models or b) the median of the KGE's calculated from each bootstrapped model simulation output

individually. From the text, it seems a) is done, but figure 3 suggests b). Please clarify. There are actually several points in the text where this is not completely clear (e.g. also 528), and I think it is relevant to know.

12) 457 -459: I don't think correspondence with KGE or any other metric should be used as an argument for using NMI, because then why not just use KGE as similarity metric? Ideally, arguments for suitability should come from some desiderata for the metric, based on what you set out to measure.

13) 465-468: I think another reason for decreasing entropy is the negative feedback of decreasing evaporation with decreasing storage, which stabilizes the system and drives it to some equilibrium. Any changes in storage tend to dampen out over time because of this mechanism.

14) 478-479: See point 6/7.

15) 596: Given points 6/7 This interpretation should be checked after comparing against the max given by the log(nr of bins).

16) 620-623: When using mutual information as a basis for clustering, it means that hillslopes outputs within one cluster are predictable from each other, not per se that they are similar. This may be another explanation for the large differences between the bootstraps.

Specific minor comments:

15 : Implies -> assumes?

144-145 either 0 or 2 commas

148: what is meant by "free water"?

184: I would suggest trying to slightly rephrase this to make it clearer of what you calculate the entropy. Maybe adding "distribution of outputs". Shannon entropy of the ensemble sounds too vague.

188-190: this is some repetition of 110-114. I agree it is worth emphasizing, but maybe rephrase and change "state" to "reiterate"

304: Bit = abbreviated from "binary digit"

310-311: Be more specific of what the Shannon entropy is calculated. This needs to be a probability distribution.

387: Maximum mutual info depends on the minimum of two entropies rather than the joint entropy.

392: Scaling with the maximum of the 2 entropies rather than the minimum seems somewhat strange, as it is actually bounded by the minimum. On the other hand, it can still be interpreted as giving the proportion of predictable behavior in the higher entropy variable, given the lower entropy one. This is however, strongly influenced by the choice of binning, while scaling with the min(H) would be less sensitive to that. I would say it is worth discussing the choice briefly, as I think it is more conventional to use the min(H) for normalizing.

395-396: Also without the normalization, MI is symmetric. Before and after the normalization, it does not satisfy the axioms of a distance function (specifically, it does not satisfy the triangle inequality).

Figure 3: bottom panel legend: "Compressed models" plural would be better fitting description of 1st line. Vertical axis should be labeled "cumulative frequency"

503: NMI efficiency: I would recommend not to use this term (first appearance in manuscript), because efficiency suggests a positively oriented score: higher=better, like NSE. For 1-NMI, lower is better.

516: consistently -> consistent?

534: I don't think a factor 17.5 would define as 2 orders of magnitude. You can leave out the last sentence as it is redundant.

540 tow -> two

Figure 6: It would be helpful to also indicate the KGE for the full model (area weighted mean of the hillslopes). I am not sure if this is anywhere in the paper. But that seems to be a good upper benchmark.

560: "information content of the hillslope responses" sounds too vague, too loosely defined

562: Similarity is here defined as having the same value, but many readers would associate similarity of hillslope response with moving in a similar direction / pattern. Please add some words of context to clarify this.

563: What is meant with the absolute minima and local maxima? In what?

633-634: high spatial organization -> high spatial complexity (I think the current wording is the opposite of what you want to say).

655: I think "without the necessity of normalization" is overselling the approach somewhat, because you still have to deal with two discretizations that determine how your bits of entropy in 2 variables compare to each other.

I would add in paragraph 49 some statement about the relevance of the nr. of data points in the choice of binning.

673: what is meant with "the other way around" ?

725: maybe add the exact equations used for easier reproducibility?

729: equally distant -> equidistant

Figure 8: in panel B, the fact that NMI=0.65 for a perfect relation is illustrative of the dependence on binning, that is introduced by normalizing for max(H) rather than min(H).

Reference:

Mutzner, R., Tarolli, P., Sofia, G., Parlange, M. B., and Rinaldo, A. (2016) Field study on

drainage densities and rescaled width functions in a high‐altitude alpine catchment. Hydrol. Process., 30: 2138–2152. doi: 10.1002/hyp.10783.

---

## Referee Comment (RC2) · C. Harman (Referee) · 30 Mar 2018

Entropy over hillslopes or time?

This paper addresses a very interesting idea and is very well written and presented. I have only one major concern, but it is right at the core of the matter and addressing it may affect the interpretation of the results.

The main issue has to do with the calculation of entropy. My question, succinctly, is whether $x$ in equation 2 refers to hillslopes or to timesteps? Or sometimes one, and sometimes the other? I have scoured the manuscript and find contradictory statements that could lead to both conclusions. The conflation between these seems to me to cast doubt on the assertion in equations 4 and 8 that the entropy can be used to determine the number of functional groups necessary to model the catchment in a non-redundant way, and that results suggest the ideal number is time-varying - major conclusions of

the paper.

The statement in Line 311 that Shannon entropy is calculated for each timestep suggests that it is being summed over hillslopes (one Shannon entropy for each timestep). That is, $p(x)$ is the marginal probability that a hillslope has a particular discharge at that timestep, and so $p(x)$ must be normalized such that the total probability at each timestep is 1. This is the entropy that is presumably being plotted in figure 4c and d.

However, if this is the case, the entropy would be maximized at any given timestep if the discharge from a randomly chosen hillslope has an equal probability of occupying any particular discharge bin. This conflicts with the definition given in equation 4. Wouldn't the maximum entropy be $\log_2(M)$, where $M$ is the number of bins? Given that you have log-spaced bins starting at 0.01 and increasing by 8.5% each time up to a maximum of around 0.7 mm/hr, it looks to me like you have about 26 bins. That would give a maximum entropy of about 4.7 bits.

It seems important that 4.7 bits is slightly larger than the largest the entropy values you observe in Figure 4c.

However, there are other points in the paper that suggest this interpretation cannot be the case (at least not all the time). What would be the meaning of $y$ in equations 3,5, 6  7? It would have to be the set of discharges at a different timestep from $x$. But then all the calculations would be about the mutual information between timesteps, and that is not what this paper is about. It seems then that $x$ and $y$ are now different hillslopes (one Shannon entropy for each hillslope). Now $p(x)$ is the marginal probability that a timestep has a particular discharge at in a given hillslope. Also $p(x)$ must be normalized such that the total probability across all timesteps for each hillslope is 1 (rather than across all hillslopes for each timestep). The $p$ associated with a particular hillslope/timestep pair will differ depending on which marginal distribution it is being considered a part of.

This is the only way I can understand how it is possible to plot both a time-varying

entropy in figure 4c, but also have a single invariant normalized mutual information (NMI) metric for each hillslope for use in the functional classification. It seems different marginal distributions are in play at different points in the manuscript, and they are not clearly distinguished.

I find myself doubtful that Equation 4 and 8 hold. Equation 4 does not seem to me to have the meaning implied, for the reasons I give above - maximum entropy occurs when all discharges are equally likely and has nothing to do with the number of hillslopes. Similarly, the entropy of a coin-toss experiment is maximized if the coin is fair, and has equal likelihood of landing on either side, and has nothing to do with the number of times I flip it.

This doubt then extends to equation 8. I find myself uncertain as to whether it is valid to interpret the entropy in figure 4c as saying anything about the time-varying number of non-redundant hillslopes required to simulate the result. If the theoretical maximum is indeed 4.7 bits, does that mean we know, even before we run the model, that we will need at most 26 non-redundant hillslopes? Is this how the uncertainty in the discharge immediately limits the required model complexity?

I wonder if a simple thought experiment (or numerical experiment) would help illustrate the validity of all this? One where we know what the 'true' number of non-redundant 'hillslopes' is, and it can be shown that this is recovered by the analysis?

---

## Author Comment (AC1) · 26 Apr 2018

Reply to Referee #1 Steven Weijs:

***Steve Weijs (SV): Summary and Recommendation:*** *"Generally, the paper is well-written, with interesting thoughts put forward in the discussion. I found the idea very relevant, and think there is a large potential for further work in this direction, perhaps focusing also on predictability rather than mostly on similarity. For the current paper, I have a number of comments/questions that I think would need some clarification in the paper. The comments are itemized below. I think most of the suggestions could easily be addressed/clarified in this paper, while some points may be more suitable to be addressed in further work"*

**Ralf Loritz (RL):** We would like to thank Steven Weijs for his comments, especially as his work influenced and inspired ours. In the revised manuscript we will follow many of the reviewer's recommendations, because this will definitely improve our study. Furthermore, do we see from the comments that some parts of our study need better or more detailed explanations which we aim to provide in the revised version of our manuscript.

**General comments**

***1. SV line 75***: *It is mentioned that some of the energy is dissipated in runoff concentration, and the remainder is exported as kinetic energy of streamflow. I would say that the vast majority of the energy is dissipated and only a tiny remaining fraction exported. If I am not mistaken, a back of the envelope calculation with reasonable (conservative) numbers : average height of catchment above river = 100 m, 100% runoff ratio, and 2 m/s flow velocity, gives me that 99.8% of energy is not exported as kinetic energy by stream flow. To dissipate a smaller portion, we would have to build large constructions (dams, penstocks) to get more kinetic energy and utilize for hydropower, and that is usually mainly saving the in stream dissipation, not much of the dissipation from runoff concentration.*

**RL:** Very valuable point. Overland flow is in fact highly dissipative and we will stress in the revised manuscript that the specific kinetic energy export per unit volume surface runoff is small. For completeness it is worth to mention that water which infiltrates into the soil changes its energy state by adding potential energy to the system, but at the same time reduces capillary binding energy. We will rephrase the sentence to: "The majority of this available energy is dissipated during runoff concentration, while the remaining part is exported from the catchment as the kinetic energy of streamflow (Kleidon et al., 2013)".

**2. SV line 154**: *I would say that in hydrology, next to resistances, there is a large influence of storages (or capacitors in an electronics analogy) in the system that would make the dynamic behaviour unique/ different between two cases with the same driver/resistance combination. Could it be that the capacitances have some relation with the resistances (e.g. through porosity and path length through soil), that explains part of the similarity?*

**RL:** We agree for the case of matrix flow. Here the resistance term (in fact the inverse of the hydraulic conductivity) and the capacity to store water against gravity are connected through the soilwater-retention and soilwater-conductivity curve. We will change the revised manuscript accordingly.

**3. SV line 222**: *Could you briefly state what criteria were used for the subdivision in hillslopes? I would say that could be quite relevant to understand possible redundancy/similarity. Is the stream network based on an algorithm or on direct observation in the field? (see e.g. Mutzner et al, 2016).*

**RL:** The stream network and the hillslopes were delineated based on a digital elevation model with a classical hydrological terrain analysis algorithm (r.watershed) in GRASS GIS (each stream segment has two corresponding hillslopes). This approach generates a stream network after the user sets a threshold for the minimum size of an exterior watershed basin. Identifying this value without any additional knowledge about the stream network is indeed difficult. We hence varied this threshold across a range of values and tried to reproduce by visual inspection an official stream network map which was available from the Luxembourg Institute of Technology (LIST). This stream network was also derived from a digital elevation model, however was furthermore also corrected based on local expert knowledge, field observations as well as on aerial photographs. We agree that there is much room for improvement and refinement here. Thanks for pointing us at the work of *Mutzner et al, 2016. W*e were not aware of their work and will give their study a closer look next time before we discretize our landscape. However, while we agree that the specific results of this manuscript clearly depend on the discretization of the landscape, the overall message of the manuscript does not. Nevertheless, we again agree that it would be an interesting question to see for instance how sensitive the proposed approach is (as well as the hydrological model) on different discretizations schemes. We will rephrase this section and give some extra details how the landscape discretizations was performed.

**4. SV line 328**: *Apart from the precision of the data, another very important consideration in the number of bins chosen is how the number of bins compares to the number of data points (this is the same consideration that goes into choice of bins when presenting data in a histogram). If bins are too fine for a relatively low number of data points, this will result in values always close to log(m), where m is the number of data points. If too coarse, we lose information that was in the distribution. It would be helpful to mention the number of bins used for both variables, and the number of data points, to clarify that the two are in balance. Perhaps also some discussion is warranted about using constant bins over the year vs. rescaling bins every time step using the maximum model. As noted, this choice basically reflects the question of which we quantify the information contained in the answers to it.*

**6. SV line 352:** *The upper bound is not only related to the number of models, but also to the number of bins used to discretize the values. Now, if I understand well, the number of bins is unbounded, but in practice, there is only a maximum number of bins occupied (would be good to know how many). Depending on the range of the bins, in order to reach the maximum entropy, the models may need to simulate unrealistically high or low values. Therefore I have some problem with the interpretation of log(N) as the maximum attainable entropy / state of zero spatial organization.*

**RL:** You are right, from a practical viewpoint the number of occupied bins is limited. The idea behind log(N) (state of zero organization) is in this sense to provide a theoretical upper limit, even if our model is unable to reach this value (see also the detailed answer to Ciaran Harman).

The basic idea behind the entire approach is that as soon as two models fall in the same bin they are from a functional perspective indistinguishable. This is why we used the measurement accuracy as bin size, which is from a physical perspective a meaningful definition for the indistinguishability of two measured or simulated values. In our study this means that our 105 models will always produce redundancy simply because it is physically unrealistic that all 105 bins are occupied. However, we still believe that this theoretical perspective has some value. Simply by using a different model realization, looking at a different quantity or working in a different environment we could also reach this value in practice. Furthermore, if a model is working close to this value it tells us that the chosen spatial resolution might be coarse.

We thank the referee for this valuable comment. We will update section 3.1.2 where we introduce this idea. We will furthermore as recommend add the "maximum reachable entropy" based on a physical meaningful number of bins to our manuscript (e.g. in Figure 4) and report the maximum number of occupied bins. In a final step we will add a plot to the appendix where we try different

binning sizes to show how sensitive the approach is with respect to different bin sizes. For further details see also the answer to Ciaran Harman.

**5. SV line 334:** *It is not completely clear to me if/how the soil moisture probe uncertainty is used in the calculation of the bin width. Is it a relevant factor in the bin width if observations are not directly used to compare with the simulations?*

**RL:** The Colpach catchment is highly instrumented with a wide range of different sensors. Among others with 46 5TE soil moisture probes (for further details see Loritz et al. (2017); https://www.metergroup.com/environment/products/ech2o-5te-electrical-conductivity/). The manufactures of these soil moisture sensors state that the measurement accuracy of these sensors is around 1% relative water content. The soil layers in our models are 1m thick and we use a saturated water content of 0.57 %. This means that per $m^2$ we can at maximum fit 570 liter into our soil layer. We now simple transfer the error of our soil moisture observations to our storage simulations. We use a constant step size of 10mm (1%) with bins ranging from 10 mm (1%) to 570mm (57%) (maximum saturation of our soil). We think that this is a physical meaningful bin size as it represents the minimum measurement accuracy. However, we agree that we could have also come up with a different bin size depending on the question we would like to ask. We apologize that this was not clear in our manuscript in the first place and will add a sentence in the revised manuscript. Furthermore will we show in the appendix how different binning sizes change the Shannon entropy.

**7. SV line 348-350**: *I think there is also another key issue here: predictability from other hillslopes is not the same as giving identical unit runoff values. If e.g. runoff follows the same pattern with a simple scaling factor, a time shift, or any one-to-one relation, I would say the hillslopes are redundant to some extent. Giving identical runoff values is a sufficient, but not a necessary condition for providing information that is redundant.*

**9. SV line 357:** *precision: I am not sure if this statement is accurate, see point 7*

**RL:** Thank you for this comment. We agree that in general one system might be perfectly predictable from the other, even if the values fall not into the same "bin". However, as soon as you are aware of a scaling factor or any other transfer function you could still use our approach. For instance, one could account for a scaling factor using the differences of the discharge values between the time steps instead of the absolute runoff values. In this case two hillslopes would be redundant if they show the same dynamics and not necessarily the same discharge value. Something similar could be

done a-priori if we know or assume that time series are shifted in time. In the end it depends on the question and on data availability during model execution how one defines redundancy or similarity with respect to an observed or simulated value.

**8. SV line 356:** *but attaining this maximum would also mean the rain has to be white noise spatially*

**RL:** We agree that an apparent spatial covariance of rainfall will create redundancy in the simulation ensemble and thus reduce the entropy below the maximum. We will stress this in the revised manuscript.

**10. SV line 359**: *identical : I think spatial organization/compressibility is more related to how predictable one hillslope output is from the other, rather than how identical they are, see also point 7*

**RL:** Good point. With identical we mean here "functionally identical" rather than structurally identical. We will rephrase this sentence to:

"At the other end of the spectrum, one may have a state of perfect spatial organization in which all 105 hillslope models are within the error margin of observations perfectly predictable form each other."

Yet we think that predictability and similarity are very closely related. We think that structural similarity, i.e. similar controls of gradients and resistance terms implies functional similarity, in case these are forced by similar forcing and start at a similar state. This implies that structurally similar systems are mutually predictable.

**11. SV line 442:** *The median of all realizations is not a compressed model, since it would need calculation results from all catchments. Please clarify whether you calculated: a) the KGE of the median of the simulation outputs from the 1000 bootstrapped models or b) the median of the KGE's calculated from each bootstrapped model simulation output individually. From the text, it seems a) is done, but figure 3 suggests b). Please clarify. There are actually several points in the text where this is not completely clear (e.g. also 528), and I think it is relevant to know.*

**RL:** We apologize that this was not clear in the first place. We calculated the KGE for all 1000 randomly generated model realizations separately (the distribution in Figure 3 and 6) as well as the median KGE from all these realizations (the dotted line in Figure 3 and 6). You are hence right to

calculate the median KGE we need to run all model combinations. We discuss this point shortly in the same section where we argue that an approach to aggregate the models similar to the one shown in our own work (Loritz et al. 2017) might be the better avenue to come up with a spatially aggregated catchment model. However to keep the manuscript short and too improve the readability we chose this rather ad hoc bootstrap method. We are sorry that this was not clear and will rephrase the above mentioned sections in the revised manuscript.

**12. SV line 457 -459:** *I don't think correspondence with KGE or any other metric should be used as an argument for using NMI, because then why not just use KGE as similarity metric? Ideally, arguments for suitability should come from some desiderata for the metric, based on what you set out to measure.*

**RL:** The idea behind the comparison of the KGE and NMI was that Hydrologist are familiar with the KGE but not necessarily with the NMI. We wanted to relate the NMI to a known quantity which has proven to be sensitive to capture differences between stream flow time series. If the Editor wishes we can remove this comparison, however we still think it could be valuable for some readers which are not familiar with the NMI.

**13. SV line 465-468:** *I think another reason for decreasing entropy is the negative feedback of decreasing evaporation with decreasing storage, which stabilizes the system and drives it to some equilibrium. Any changes in storage tend to dampen out over time because of this mechanism.*

**RL:** Good point, we will try to add this to the corresponding section in the revised manuscript.

**14. SV line 478-479**: *See point 6/7.*

**15. SV line 596**: *Given points 6/7 This interpretation should be checked after comparing against the max given by the log(nr of bins).*

**RL:** Thank you for these comments. We hope that these points become clear in a revised manuscript and further hope that the answer to point 6, 7 and to Ciaran Harman are satisfying.

**16. SV line 620-623**: When using mutual information as a basis for clustering, it means that hillslopes outputs within one cluster are predictable from each other, not per se that they are similar. This may be another explanation for the large differences between the bootstraps.

**RL:** Good point. We would say they are at least partly functionally similar but not necessarily structurally similar. In our virtual environment, where we kept all aspects except geometry identical across the hillslopes, this is to a certain extent the same; in the real world not necessarily.

**Technical / minor comments**

*SV Line 15:* changed

*SV Line 144-145:* changed

*SV Line 148:* removed

*SV Line 188-190:* changed

*SV Line 310-311:* changed

*SV Line 387:* We think that your and our statement is correct here*.*

*SV Line 392:* We will add a sentence as well as showed an example in the appendix in a revised manuscript.

*SV Line 395-396:* We agree with the referee and will rephrase this part*.*

*Figure 3:* changed

*SV Line 503:* Removed the term efficiency

*SV Line 516:* changed

*SV Line 534:* Sentence removed.

*SV Line 540:* changed

*Figure 6:* We added a sentence to our results section.

*SV Line 560:* We will rephrase this sentence.

*SV Line 562:* We will add a sentence.

*SV Line 563:* removed

*SV Line 633-634:* changed

*SV Line 655:* Agreed. We removed the part "without the necessity of normalization".

***SV Line 655:*** *"I would add in paragraph 49 some statement about the relevance of the nr. of data points in the choice of binning.*

We agree with the reviewer that the point he raises is critical when estimating the Shannon entropy. However, we think we discuss this point already in section 3.1.1 right at the start of our method section. We also cite Gong et al. 2014 and Pechlivanidis et al., 2016 as reference for further information in this respect.

***SV Line 673:*** removed

***SV Line 725:*** Good idea we will do that.

***SV Line 729:*** changed

***SV Figure 8:*** We added a sentence in the Appendix. Thanks for this notice.

---

## Author Comment (AC2) · 26 Apr 2018

Reply to Referee #2 Ciaran Harman:

**Ciaran Harman (CV): Summary and Recommendation: "***This paper addresses a very interesting idea and is very well written and presented. I have only one major concern, but it is right at the core of the matter and addressing it may affect the interpretation of the results.*"

**Ralf Loritz (RL)***:* We would like to thank the second referee for the time and the effort he put into writing his review. The points he raises are relevant and addressing them will help improving our manuscript. We hope that after this discussion (as well as after we revised our manuscript) all issues he raises can be clarified.

*CV: The main issue has to do with the calculation of entropy. My question, succinctly, is whether in equation 2 refers to hillslopes or to time steps? Or sometimes one, and sometimes the other? I have scoured the manuscript and find contradictory statements that could lead to both conclusions. The conflation between these seems to me to cast doubt on the assertion in equations 4 and 8 that the entropy can be used to determine the number of functional groups necessary to model the catchment in a non-redundant way, and that results suggest the ideal number is time-varying - major conclusions of the paper.*

*However, there are other points in the paper that suggest this interpretation cannot be the case (at least not all the time). What would be the meaning of y in equations 3,5, 6  7? It would have to be the set of discharges at a different timestep from x. But then all the calculations would be about the mutual information between timesteps, and that is not what this paper is about. It seems then that x and y are now different hillslopes (one Shannon entropy for each hillslope). Now p (x) is the marginal probability that a timestep has a particular discharge at in a given hillslope. Also p(x) must be normalized such that the total probability across all timesteps for each hillslope is 1 (rather than across all hillslopes for each timestep). The p associated with a particular hillslope/timestep pair will differ depending on which marginal distribution it is being considered a part of.*

**RL:** This is an important point and indeed the manuscript is separated into two parts. The first part deals with the identification of the required spatial complexity of a hydrological model (in our study this is expressed by the number of hillslopes). Here, we use the Shannon entropy (equation 2) as well as the concepts of the minimum number of questions (maximum compressibility; equation 8) to analyze how much redundancy with respect to storage or runoff we produce with our hillslope models (redundancy here means that two or more hillslopes return, at a given time step, discharge

or storage values falling into the same bin). We calculate the entropy of the discharge ensemble (= set of discharge values from all hillslopes) for each time step and show that they are partly redundant. In addition, when looking at the time series of entropies, we see that the degree of redundancy varies in time. While we agree that the magnitude of Shannon entropy of the discharge simulations at a given time step does not necessarily related to dynamic similarity in general, it is indeed the case for the special conditions of our virtual experiment: Here the hillslopes differ only with respect to topography, all further parameters and the forcing are equal. In that special case, we can establish a direct link between cause (similarity of topography) and effect (redundancy of discharge).

The main conclusion of the first part of our study is that the model setup is in general compressible, and that the degree of compressibility varies with time. In the second part we make use of the first finding, but for the sake of brevity ignore the second. This means that we simply took the mean of the time-series of Shannon entropies from part one, which expresses the mean compressibility of the model over time. Clearly, for a particular timestep this may be an over- or underestimation of the true compressibility. We discuss the shortcoming that the spatial resolution of a compressed catchment model should also vary in time in detail in section 5.2.

However, to avoid confusion and to make this point clear we will add a section at the end of our objectives where we clearly explain the two main research questions of our manuscript.

**CV:** *However, if this is the case, the entropy would be maximized at any given timestep if the discharge from a randomly chosen hillslope has an equal probability of occupying any particular discharge bin. This conflicts with the definition given in equation 4. Wouldn't the maximum entropy be log(M) , where M is the number of bins? Given that you have log-spaced bins starting at 0.01 and increasing by 8.5% each time up to a maximum of around 0.7 mm/hr, it looks to me like you have about 26 bins. That would give a maximum entropy of about 4.7 bits. It seems important that 4.7 bits is slightly larger than the largest the entropy values you observe in Figure 4c.*

*I find myself doubtful that Equation 4 and 8 hold.  Equation 4 does not seem to me to have the meaning implied, for the reasons I give above - maximum entropy occurs when all discharges are equally likely and has nothing to do with the number of hillslopes. Similarly, the entropy of a coin-toss experiment is maximized if the coin is fair, and has equal likelihood of landing on either side, and has nothing to do with the number of times I flip it.*

*This doubt then extends to equation 8. I find myself uncertain as to whether it is valid to interpret the entropy in figure 4c as saying anything about the time-varying number of non-redundant hillslopes required to simulate the result. If the theoretical maximum is indeed 4.7 bits, does that mean we know, even before we run the model, that we will need at most 26 non-redundant hillslopes? Is this how the uncertainty in the discharge immediately limits the required model complexity?*

**RL:** Thanks for this point. Also the second referee made this point which highlights that we need to address this issue in a revised manuscript. Most importantly, we must make clear that this discussion of maximum entropies contains two perspectives (experiment and system).

In statistical mechanics the maximum entropy relates to the log of the maximum number of different microstates of a system with a given energy. This leaves us with the question what determines the number of microstates in our study. We think that your interpretation of the maximum number of bins and our interpretation of the maximum number of models do not at all contradict each other but are supplementary.

Let's assume we have a fair dice with six possible outcomes. The entropy of this system is linked to the possible states the "*system*" could reach and would hence be $\log_2(6)=2.58$. Now depending on our "*experiment*" (maybe we lost our glasses) we only ask the question is the value larger or smaller than 3. In this case the maximum entropy of our "*experiment*" would with two possible states be $\log_2(2)=1$.

Following this line of thought, we argue that there is a difference between the maximum entropy of the system and of the experiment. In our manuscript the maximum information our model ensemble can produce about any given output is $\log_2(105)=6.7$ bits. If our goal is to simulate a distribution with a higher entropy our model is doomed to fail. On the other hand, as you and Mr. Weijs correctly commented: in our specific experiment, given the uncertainty in our discharge observation, it is unlikely that we reach this theoretical value as long as our models produce reasonable results. $\log_2$(No. of possible occupied bins) reflects hence a physical viewpoint of our specific experiment because we do not expect that our discharge is larger than a certain threshold. Here picking the maximum discharge value (0.75 mm/hr) seems reasonable (leads to 54 bins and a maximum entropy of 5.7). However, one could also argue that a value of 1 mm/hr is physical reasonable in this environment which would result in a number of bins around 58. Nevertheless, in our experiment it would still not lead to 105 bins meaning that our model will always produce redundancy.

As already stated in the answer to the first referee we need to make this clear in a revised manuscript and rephrase section 3.1.2 in this respect. We think that both concepts are useful if we

want to identify the needed spatial complexity of a hydrological model and we are thankful that both referees pointed out the difference between the two concepts.